# 'You Scratch My Back and I'll Scratch Yours'? Support to Academics Who Are Carers in Higher Education

**Marie-Pierre Moreau \* and Murray Robertson** 

School of Education and Social Care, Faculty of Health, Education, Medicine and Social Care,
Anglia Ruskin University, Cambridge CB1 1PT, UK; robertsm2@roehampton.ac.uk
\* Correspondence: marie-pierre.moreau@anglia.ac.uk

**Abstract:** In recent years, it has become common for individuals to juggle employment and unpaid care work. This is just as true for the England-based academic workforce, our focus in this article. We discuss how, in the context of English Higher Education, support for carers is enacted and negotiated through policies and practices of care. Our focus on academics with a diverse range of caring responsibilities is unusual insofar as the literature on care in academia is overwhelmingly concerned with parents, usually mothers. This article is informed primarily by critical and post-structuralist feminist perspectives. We draw on a corpus of 47 interviews conducted with academics representing a broad range of caring responsibilities, subjects, and positions. A thematic analysis reveals how carers' relationship with the provision and policies of care support at an institutional level is characterised by ambivalence. On the one hand, participants approve of societal and institutional policy support for carers. On the other hand, they are often reluctant to position themselves as the beneficiary of such policies, expressing instead a general preference for support from outside the workplace or for workplace-based inter-individual and informal care arrangements. This resistance is particularly noticeable in the case of participants with caring responsibilities other than the parenting of healthy, able-bodied children and of those whose gender, class, racial, or sexual identity do not conform with the figure of the 'ideal academic', contributing to their othering in the academic realm. These findings have significant implications for policies supporting carers, pointing to the need for greater visibility and recognition of caring responsibilities in academia, especially in terms of their diverse identities.

**Keywords:** carers; higher education; academics; policies

## 1. Introduction

Recent research highlights how, in the UK as in many other countries, juggling employment with caring responsibilities has become an increasingly widespread occurrence (Carers UK 2014). In English Higher Education, the focus of this article, little evidence has been systematically collected across the sector regarding the caring status of employees, academics or otherwise. While a variety of studies describe how, as in other sectors, combining employment and care work is commonplace (Moreau and Robertson 2017), extant research rarely focuses on academics who are caregivers. When it does, it usually concentrates on parenting, often mothering, with little consideration of other caring responsibilities. Additionally, the analytical lens underpinning this body of work tends to omit the intersectionality of identities and how the hyphenated identity[1] of the academic carer is compounded by social class and ethnic background, sexual orientation, disability, and gender. Furthermore, the literature

---

[1] 'Hyphenated' is used in this context to refer to the dual, often conflicting, identities of participants as academics who are also carers.

in this area often describes the relationship between paid and unpaid work in terms of 'work–life balance', with the framing of this matter in individual terms rendering invisible the relations of power at play.

Following the above, this article explores how, in the context of academic cultures which have been described as 'care-free', support for carers in academic jobs is enacted and negotiated through policies and practices of care, for example when it comes to making decisions about taking paid or unpaid leave to care for another person. Using critical and post-structuralist feminist theories, we draw on two separate, yet related, research projects: *Carers and careers: Career development and access to leadership positions among academic staff with caring responsibilities* and *'Care-free at the top'? Exploring the experiences of senior academic staff who are caregivers*, respectively funded by the Leadership Foundation for Higher Education, now Advance HE (Moreau and Robertson 2017) and by the Society for Research into Higher Education (SRHE) (Moreau and Robertson 2019). These two projects generated 47 semi-structured interviews with academics working in a range of positions and institutions across England.

This article opens with a presentation of our theoretical framework and methodology, before turning to an analysis of participants' narratives. We first discuss their views of the state and institutional support offered to carers in academia—a legitimate area for policy intervention for most—before discussing their views of formalised care support vs. individualised arrangements. We then proceed to discuss the hierarchies and intersectionalities of care/rs and how this frames their recourse to such policies, before shedding light on the apparent contradiction between academic carers' views of institutional support and their simultaneous reluctance to use formal institutional policies. We conclude with a discussion of the findings and of their implications for carers and higher education institutions.

## 2. Theoretical and Methodological Framework

### 2.1. Theory

This article is informed by critical and poststructuralist feminist theories. It acknowledges the centrality of power relations such as gender on individual lives and societies at large, including in relation to doing academic and care work—two highly gendered activities (Moreau 2016). Consistent with a post-structuralist perspective, access to a positional identity as an academic and a carer is conceptualised as framed by intersectional and shifting power relationships, which operate within discourses of care and academic work. These discourses are being simultaneously negotiated by individuals, as they navigate the tensions between 'doing care' and 'doing academic work'. Linked to the long-lasting, well-evidenced opposition between academic and care work (Fraser and Gordon 1997; Grummell et al. 2009; Leathwood and Read 2009; Lynch 2010; Lynch et al. 2009) and to the association of men with the former and women with the latter (Crompton et al. 2007), the relationship of carers, particularly of women carers, with academia tends to be fraught with tensions (Moreau 2016; Moreau and Robertson 2017, 2019). This is particularly the case in societal contexts where, as discussed elsewhere in more depth (Moreau 2018), a liberal and individualist conception of the welfare state prevails, which constructs the combining of paid and care work as a matter of private responsibility (Faucher-King and Le Galès 2010). In contemporary times, performing an identity as an academic and a carer takes place against a background in which academia and the family have been described as 'greedy institutions' (Coser 1974) which require full availability and loyalty. The demands on academic carers are also likely to be exacerbated by two discourses which we have described at length elsewhere: a discourse of intensive parenting (in reality, intensive mothering, in the context of the cultural association of femininity with care work) and a discourse of the neoliberal university (Moreau 2016).

Thus, we acknowledge individuals' negotiations of what constitutes care, favouring a definition of care as something we 'do' or 'perform' rather than 'are', paralleling Butler's (1990) conceptualisation of gender. Care is also constructed as inherently relational. While we acknowledge that some individuals

have significant caring responsibilities that they cannot easily renounce, an implication of a relational conception of care is that people are all embroiled in reciprocal and non-reciprocal relationships of care, as care-giver and care-receiver. Indeed, as argued by Barnes:

> Life histories demonstrate that caregivers may also be care receivers—at different stages of their life, or at the same time—and that the categorical distinction between carer and 'dependant' may not stand up. This applies in situations where people with learning disabilities become parents . . . when spouses care for each other, when a disabled mother gives birth to a disabled child or when elderly parents provide as well as receive care from adult children. (Barnes 2011, p. 172)

Last, influenced by authors who have written about care ethics, we acknowledge that care work is multifaceted, with some aspects of care (e.g., organisational or affective) which cannot be easily commodified and delegated to others (Lynch et al. 2009).

*2.2. Methodology*

This article is informed by a corpus of data collected as part of two related research projects: *Carers and careers: Career development and access to leadership positions among academic staff with caring responsibilities*, funded by the Leadership Foundation for Higher Education, now Advance HE (Moreau and Robertson 2017) and *'Care-free at the top'? Exploring the experiences of senior academic staff who are caregivers*, funded by the SRHE (Moreau and Robertson 2019). Both projects shared a similar theoretical framework and, to some extent, methodology. A key finding of the first project was that the most senior levels of academia appeared relatively 'care-free', leading to the second project, which had a more specific focus on those senior academics. Thus, both projects were closely linked, with the second project conceived as a follow-up to the first one.

As part of the first project, three institutional case studies were identified so as to gain an in-depth and contextual understanding of the lives of participants. The three case studies were: a pre-1992 Russell Group institution[2] based in the north of England; a post-1992 institution based in the London area; and a post-1992 institution based in the south of England. Sampling was based on the contrasts these institutions provided in terms of geographical location and status. In each institution, we conducted a desk search to gain some general understanding of the institutional ethos and of carer-related provision and policies. We then conducted interviews with one to two members of staff in a relevant policy role (e.g., HR or diversity and equality, referred to hereafter as 'policy staff') and interviewed eight to nine academics with caring responsibilities. However, interviews with policy staff are not considered in this article, which solely explores the views of academics.[3] Interviews with academic staff focused on their social and educational backgrounds; career trajectory; family and caring circumstances; general experience of being an academic and a carer; relationship between care work and career development; awareness of care support provision in place; views on institutional policies and practices; main source of support; responsibility of the university in supporting carers; and other aspects they identified as worthwhile discussing.

In each case study, access to interviewees was negotiated with a member of staff with responsibility for equality and diversity. They or a representative circulated a call for participants. Interviews were conducted face-to-face (on campus) or by phone or video conferencing (using the Skype interface). The diversity of the academic sample was a prime concern, as we aimed to look at caring responsibilities from an intersectional perspective and to draw comparison across groups. As a result, we closely monitored the recruitment of participants and circulated the call several times. We also used additional recruitment strategies (e.g., asking interviewees to recommend other participants and targeting

---

[2]　In the UK, post-1992 universities are institutions which have gained university status that year, while pre-1992 have had this status for longer periods of time. Russell group universities are a group of research-intensive UK-based institutions.

[3]　For a discussion of policy staff's views of care/rs, see Moreau and Robertson (2017).

specifically those from under-represented groups) to increase the numbers of male and minority ethnic participants and to ensure diversity in relation to the nature of their caring responsibilities and to the job position. In total, as part of this first project, we conducted 27 interviews with academics.

All interviews were audio-recorded and professionally transcribed. The transcripts were imported into an NVivo database and subjected to a thematic analysis (Robson 1993), with key themes derived from the original research questions, from the interview questions, and from the repeated readings of the transcripts. A coding grid was designed, with data coded under 'coding nodes' (see coding grid and interview schedules in Moreau and Robertson 2017). Thematic reports were produced for each node, with the key findings summarised. Comparisons were also drawn between different groups of participants, e.g., considering differences related to gender, the nature of caring responsibilities and the position.

As noted above, the second project adopted a broadly similar methodology and theoretical framework with, however, some differences justified by some of the issues experienced as part of the first project. Due to the difficulties we had faced in the first project in recruiting senior academics and to the small numbers involved, we decided not to use an institutional case study approach. Instead, a call was broadly circulated through professional networks and organisations. This recruitment strategy was also deemed more suitable on an ethical level due to the small numbers of staff in senior positions. The recruitment of volunteers was monitored to encourage maximum diversity, particularly in relation to gender, ethnicity, position, subject and institution, all of which have been shown to affect the production of academic identities (Clegg 2008; Deem 2003). Yet, similar in this to the first project, recruiting male and Black and Minority Ethnic (BME) academics of any gender group proved challenging. Our data suggest that the under-representation of male academics in studies about care may reflect their greater reluctance to take up a carer identity. The small numbers of BME academics involved in this research may be linked to their relative exclusion from academia, although further research would be needed before some conclusions can be drawn. Interviews were audio-recorded and professionally transcribed. However, in contrast with the first project, we decided not to use the NVivo qualitative data analysis software. Instead we produced some structured summary of each transcript. This enabled us to retain the wholeness of each narrative, while the identification of themes structuring each summary allowed us to draw comparisons between interviews with specific attention given to differences relating to the position, and to gender and its intersections with other identity markers. In total, 20 senior academics were interviewed as part of the second project.

Altogether, these two projects generated a corpus of 47 semi-structured interviews with academics, including 31 women and 16 men. Ages ranged from 31 to 66 years old. The positions held were diverse, including for example research assistant, lecturer, head of department, professor, pro vice-chancellor and vice-chancellor, with some interviewees holding multiple roles. A range of subject backgrounds was represented. Twenty-nine interviewees self-identified as White British, 15 as White Irish, White Jewish or from another White background considered as 'White Other' in the UK census, two as belonging to a Black and Minority Ethnic group (with further detail retained to protect anonymity) and one as Mixed Race. Their family circumstances were highly diverse and included living on their own, with a partner only, with a partner and one or several children, and with a partner and other adults.

Both projects underpinning this article adopted a broad exploratory angle, although the second project focused on a more narrowly defined group. This is consistent with our epistemological and theoretical positioning, and with the fact that care/rs in academia has attracted little research interest thus far. Also consistent with this approach, being 'an academic' and 'a carer' was based on self-identification rather than on contractual or other pre-established definitions. In doing so, we acknowledge that definitions of care/rs are multiple, dynamic and, ultimately, contested (Tronto and Fisher 1990). The complexities and diversities of care are maybe most strikingly illustrated by the circumstances of those who volunteered to take part in the interview. Some were caring for a child, a grandchild and/or an elderly parent. Some cared for a relative, partner or friend experiencing a disability, a long-term ill-health or old age. Some were caring for multiple individuals at the same time or at different period of their lives (the so called 'sandwich generation', Miller 1981, or 'sandwich carers', Carers UK 2012).

Care work evolved over the lifecycle, both in terms of who it was provided too and how. Caregiving was sometimes occasional, sometimes regular; it covered short and long periods of time; it was provided from a proximity or from a distance; it was extensive or parsimonious; it was of a mostly practical or emotional nature. The negotiations involved in claiming a positional identity as a carer were also illustrated by potential participants who contacted the research team, enquiring whether they may qualify as carers (see example in Moreau and Robertson 2017).

## 3. Support to Carers: Formal Institutional Policies vs. Individualised Practices

### 3.1. Societal and Institutional Support to Carers in Academia: A Legitimate Area for Policy Intervention

The academics and the professional members of staff interviewed as part of the two projects all agreed that social and institutional support to carers in academia is fully legitimate. In their narratives, care was constructed as a collective and public matter. While an individualised discourse of 'risk and responsibility' (Beck 1992) was also drawn upon in some narratives, this discourse of care as a collective and public matter prevailed. In relation to state support, Pauline (Professor, caring for children and elderly parents), for example, argued that:

> . . . this is a societal thing, I mean we do not live in Britain in a society that really recognises the need for caring generally. It's very much geared up to you as the worker and how much any company can extract from you as a worker and so the other people around you, it's a complicated situation. I don't know any other country that's really got it perfect but in Britain I think we're very much in denial, especially about care for the elderly which is becoming more of a crisis because of course we have an aging population, there's more people living longer. It's becoming more of an issue.

In relation to institutional support, Jevan (Demonstrator and Lecturer, one child) commented:

> I don't think I'm necessarily supported through this other than any informal agreements I make . . . and that's very much just me going, "Oh well, can I do this, at this point, this point?" and other people going, "Yeah that's fine" so I suppose that's support, but accommodation is a better word, because it's more, you have your requirements what they want you to do and other people accommodate that or they don't. And, if they can't accommodate then you have to shift yourself around that.

Comments from Pauline and Jevan (all names are pseudonyms) are typical of our participants' views of societal and institutional support to carers. All stressed the need for, and often lack of, support, both in relation to the Welfare state (in the form of national family and care policies) and to individual institutions, with the state and the employer constructed as bearing some obligation in supporting employees with caregiving responsibilities. Social justice and business arguments were both invoked to justify a policy intervention.

### 3.2. Formal Policy Support vs. Individualised Arrangements

As observed above, support to carers through national and institutional policies is constructed by participants to our studies as legitimate. Yet, asked about sources of support when confronted with the tensions likely to arise from their dual status, the majority of participants explained that they would usually turn to specific individuals with whom they had a pre-existing relationship. Some of this support came from outside work, for example from the 'hidden solidarities' provided by friendship networks (Spencer and Pahl 2006) or by a partner (Arksey 2002). Some drew on bought-in support, for example, childcare or elderly care. In the workplace, some support was of a formal nature, for example when academics had applied to benefit from specific institutional or statutory provision, e.g., going on paid or unpaid leave (usually maternity or paternity leave), requesting flexible working arrangements or switching between part-time and full-time work. However, formal policies were often

met with scepticism when it came to their practicalities and efficiency, with the exception of parental leave (a point we come back to later).

In the workplace, support was presented as coming primarily from peers:

> Work-related problems, in the main, I've got some very close colleagues and I would say that that peer support is very effective . . . . The reason I came to [University] was because of an existing connection in [Subject] at that time and that member of staff has been a constant sort of big brother support for me. (Alasdair, Senior Lecturer, children and elderly parents)

> I would say that in my immediate circle of colleagues, that I've found quite a lot of good recognition and understanding of those responsibilities and support, so I would say that my colleagues are very supportive, and we support each other. There's a kind of recognition that people do have those [caring] responsibilities. (Gemma, Lecturer, children)

Alasdair's and Gemma's comments illustrate this 'preference' for informal, collegial support, usually of an emotional or practical nature. This view that '*the department is fine, it's the wider university organisation that's the problem*' (Isabella, Lecturer, elderly father) was widespread among our participants. It points to a strong discursive binary across participants' narratives which opposes central, institutional policy making with a more decentralised level of policy making constructed as the site of informal practices and deemed by many more supportive. This preference also has implications in relation to the visibility of care work—a point to which we come back in the conclusion.

### 3.3. Beyond 'Reciprocality': The Gendered Division of Care Work

In the narratives of participants, caring arrangements were often described as informal, inter-personal and reciprocal (Barnes 2011), and individualised (as illustrated by the quote used in this article's title and by Jevan's excerpt above). This rhetoric of reciprocality is also well encapsulated in the following quote, when Christina (Professor, children, elderly mother) notes that 'people have taken up the slack for me when I've found it difficult to get in because my kids have had problems and I take it up for other people, in the same way, because you know, it's a reciprocal thing'. However, while this support may initially appear to be both elective and equally distributed, who gives and receives support is also strongly framed by gender and other identity markers (Fraser 2016; Lynch et al. 2009). In particular, women seemed more likely than men to take up a positional identity as a carer (Acker and Armenti 2004; Amsler and Motta 2019). In the excerpt below, as in Christina's and others' narratives, the support provided between academic carers through inter-personal, individualised and informal practices is described as reciprocal. Yet it emerges that it is the 'new dad' met at a conference who is constructed as the care recipient, while the (female) interviewee and her (female) colleague are positioned as the caregivers.

> You scratch my back and I'll scratch yours, so when I now have colleagues who need to leave to look after kids, this one female colleague who has kids the same age as mine, we were both away at this conference last year and there was a new dad there and he just looked bloodshot and knackered and he slept most of the time because he was away and he had the bed to himself and he wasn't being woken up and it was just like we've come out the other side now, so it's our job to support them. They looked down on us five years ago when we were going through it, they were like 'look at them, useless women . . . ' and it's like now you're living it, just remember in five years you'll be us coming out the other side and look after the new ones coming in. (Heather, female, deputy head of department, children)

In this narrative, the doing of care work follows culturally scripted gender demarcations, while this gendering is simultaneously played down by drawing on a discourse of 'false equivalence' and reciprocality encapsulated in 'you scratch my back and I'll scratch yours'. This gendered division of care work in the workplace also echoes the gendered and patriarchal economy of care work described

by participants in relation to their 'private lives'. Christina, for example, noted how she received some support from her mother when she was a single parent and needed to attend conferences, and is now expected to look after her in a way her brothers are not. In both our studies, care work is enacted through gendered chains which are also classed and inter-generational, when female academics delegate care work to another woman and receive the support from other women in or outside their family to be able perform an academic identity, while simultaneously retaining the 'mental burden' of organising this delegation (Haicault 1984).

## 4. Hierarchies and Intersectionalities of Care/rs

So far, we have discussed how academic caregivers as a whole tend to express a preference for resorting to informal and inter-personal forms of support rather than formal policies, although we also want to acknowledge that both forms of support are linked, for example when practices are enabled or even encouraged by institutional policies or when participants renounce their right to take leave when this would have a negative incidence on their peers because no additional resources are available in their absence. However, the analysis of interviews reveals that all carers are not equal when it comes to getting support and recognition for their hyphenated identity. Those doing care work other than parenting healthy, abled children appear less likely to access formal support and, related to this, more likely to resort to individualised practices of care. Graeme (Head of School, one child) commented:

> The problem that I sometimes see is we can have a policy about maternity, all right which is clear, but then what about someone whose parents are ill other than a child. There isn't really a clear policy or relief to that. I find that paternity or maternity is quite a standard set of circumstances. There's a set start date [ ... ] These other ones are very nebulous and it's much harder to capture. It's much harder for people to articulate them, you don't know when a parent contracts x or y, it's not got such a clear rise to it.

Likewise, Catherine (Professor, elderly relatives) noted:

> It's so frustrating because I have covered for people in my department who have been on maternity leave [while she was caring for a terminally ill relative], but there's nothing for people like me.

Resorting to individualised practices of care was a widespread occurrence among those caring for a child with a disability or for an adult with a long-term illness or in their old age—precisely the types of caring responsibilities which tend to be closely associated with a sense of emotional, financial, and/or organisational struggle due to the complexity of those care needs and to the lack of recognition of these groups. A material expression of this lack of recognition can be found in the lack of formal provision available to these groups at institutional and societal level.

Like those with caring responsibilities other than parenting, those who do not fit the archetype of the 'bachelor boy' (i.e., in the UK a White British, middle-class, abled, heterosexual and 'care-free' man; Edwards 1993) were found to be the more likely to resort to individualised arrangements. There was considerable evidence of how women academics had been positioned as 'failing' academics because of their caring responsibilities, with many reporting having been at the receiving end of discriminatory and bullying practices. Ellen (Principal Lecturer, children) explained how, in her previous workplace, she had been taken off from a PhD supervision team against her will after going on maternity leave:

> I went on maternity leave for the first time and I was supervising a PhD student at the time and because I went on maternity leave I had to interrupt that. She had a couple of other supervisors so that was okay but I did want to pick it up after I came back from maternity leave. She was just about to do her Viva but I wasn't allowed to pick her up and I wasn't allowed into her Viva.

Likewise, Pauline (Professor, children and elderly parents) explained how a previous head of department positioned her as a 'bad mother', including in very explicit ways as he resorted to verbal abuse:

When I was pregnant with my son, we had a really interesting head of department who didn't think women should work if they had children [ . . . ] And he said so, loudly, several times and that was quite stressful. He would shout at me. He would tell me I would be a bad mother if I considered coming back to work after I'd had my first child.

The analysis of participants' narratives also highlighted instances when gender intersected with 'other' identities and power relations, such as sexual orientation, social class, dis/ability or ethnicity, although we acknowledge the need for further research focusing on each of these equality markers. Kat (Professor and Head of Department, care for partner) explained:

I mean I had a very negative experience at my previous university [ . . . ] Where at that point [partner's first name] was ill, but not as ill as she is now. I was on a temporary contract and so there were two or three other members of staff who were on the same temporary contract. One of them, a man, got a post elsewhere and used that to get a post at [institution]. I was told by various people that that was okay because he had a family. I pointed out actually that I did have a family and I had caring responsibilities and it felt to me at the time that somehow that wasn't recognized because she [partner] was a woman and because there weren't children. Because of that I've always been more explicit to make clear what those caring responsibilities are, so people can't hide behind a sense of, 'Oh, but it's not kids and you're not a parent.'

As a woman who is both the main carer for a same-sex partner with a chronic illness and the main breadwinner, Kat's personal and professional life does not neatly align with the prevailing, traditional arrangements associated with the male main breadwinner/female main carer model (Crompton 1999). Within heteronormative and gendered discourses of the family and paid work, she is positioned by some of her colleagues as neither a 'proper' academic nor a 'proper' carer. Yet she has access to resources which enable her to actively resist this positioning and reclaim her hyphenated identity, for example when ensuring that her colleagues are aware of her career aspirations and caring responsibilities.

## 5. Discussion: Understanding Resistance to Formal Policies and 'Preference' for Individualised Care Practices

In this article, we consider how, in the context of English HE, support to those with caring responsibilities is enacted and negotiated through policies and individualised practices of care. While combining paid and care work is now a commonplace occurrence, research on this group remains scarce, with the extant literature rarely considering caring responsibilities other than parenting. More specifically, there is an apparent contradiction between academic caregivers' construction of state and institutional support to carers as a legitimate area for policy intervention, and their reluctance, more or less exacerbated depending on their 'other' identities, to taking up the policies and provision available to them.

This reluctance to use institutional support (when it is available) and the related preference for inter-personal, individualised arrangements are likely to be compounded by several factors and linked to cultural norms at play at a societal and institutional level (Koslowski and Kadar-Satat 2019). First, reluctance needs to be related to the provision in place, with participants often commenting that these provision and policies do not meet their needs. This is particularly likely to be the case for those caring for individuals other than healthy, abled children. Even for the latter group, the increased differentiation of working conditions, including on a spatio-temporal level, means that standard provision does not necessarily meet their needs. Pauline, for example, commented:

The student numbers have just gone up so much in my working lifetime and then the number of hours we teach and that has become problematic because increasingly now three days

a week of this academic year I taught till 6:00 p.m. so I wasn't getting home until really late. The university has extended the teaching day to 7:00 p.m. now, just to try and fit in the number of students into the very small number of rooms we have and so I think for anybody with a family, even if the family is in [City], if you're teaching till 7:00 p.m., no nursery is going to be open till 7.

Second, while the spatio-temporal regimes of academic work are demanding and, as noted above, increasingly differentiated, the flexibility associated with some positions, along with the expectation that academics are independently managing their work on a spatio-temporal level (albeit under the surveillance of an institution which make them accountable), opens up the possibility to resort to informal arrangements and work around—to some extent—caring responsibilities. This flexibility and preference for interpersonal and informal arrangements dealt with at departmental or institute level were often constructed by participants as deeply embedded in university cultures:

I think in academia, because of the way it works . . . people just tend to work around things unofficially I suppose, you know, because that's what I'm doing, it's all unofficially working round things, but I don't know anything about actual policies.

(Enzo, Reader, children)

Third, the use of informal arrangements and the reluctance to apply for, say, leave unless necessary or linked to financial incentives and legal framework (as is the case for maternity leave) needs to be understood in the context of the association of academic identities with the figure of the 'bachelor boy' (Edwards 1993) which we have discussed more extensively elsewhere (Moreau 2016; Moreau and Robertson 2017). This academic figure goes back a long way. Enlightenment philosophy and the establishment of modern institutions and modern science were already underpinned by sets of binary oppositions between the body and the mind, emotions and rationality, care and academia, and ultimately femininity and masculinity (ibid.). In recent decades, discourses of the rationale academic have been re-actualised. Two new discourses appear most relevant to this argument: a discourse of intensive parenting and a discourse of neoliberal academia, with both care work and academic work constructed as bottomless tasks and care and academia as 'greedy institutions' (Coser 1974). Thus, academic cultures have changed, yet remain broadly geared towards the care-free (Lynch 2010). Even when policies targeting the needs of carers are put in place, they tend to be conceived of as an 'add on' rather than an attempt to radically transform academic institutions and normalise care, with generic policies often constructed with the figure of the 'bachelor boy' in mind (Moreau 2016).

Performing a hyphenated identity is particularly fraught for those carers who are othered and marginalised due to their other identities, e.g., female, working-class, Black and minority ethnic, disabled and LGBTQ+ academics, and those whose caring responsibilities are the most invisible and misrecognised (e.g., those caring for an elderly parent or for a relative or friend with a long-term illness). Our research is situated in societal and institutional contexts where care is constructed as an individual or inter-individual matter (from self-help to collegiality), rather than a structural issue requiring the enactment of collective solutions and a radical rethinking of the way institutions and society as a whole work (Maasen et al. 2007).[4] In this view, resorting to individualised practices is not only a practical necessity in the absence of provisions meeting the increasingly diverse needs of academic caregivers, it also becomes a way to protect a hyphenated identity at risk of being misrecognised due to the multiple layers of othering that some carers encounter. Data from both our projects show that this 'care-free' culture is prevalent at the highest levels of academia, with those belonging to marginalised groups (e.g., women) most at risk to struggle in reconciling doing care with doing senior academic work.

---

[4]　Although this is not the focus of this article, it is worth noting here that there are significant variations across and within institutions in relation to the inclusion of carers. For a more in-depth discussion of this point, see Moreau (2016).

These findings have significant implications for policies supporting carers, pointing to the need for greater visibility and recognition of caring responsibilities within academia, including in terms of their diversities and intersectionalities. Care policies must be characterised by a certain level of flexibility and developed with a broad range of intersectional identities and caring needs in mind as otherwise they risk only addressing the needs of a particular family configuration or type of 'carer'. To be effective and bring about social change, policies also need to be visible and normalised. For example, a generous and ambitious care leave policy risks not to serve its purpose if staff are not made aware of its existence before their needs arise (Moreau and Bernard 2018).

Our earlier work in this area highlights the importance of the immediate environment in producing workplaces which are inclusive to carers and other groups (Moreau 2016; Moreau and Robertson 2017). Immediate colleagues based in the same departments or research grouping are key to the production of environments which are un/supportive to carers. The line manager, in particular, was identified across participants as a key influence in terms of generating a regime of care which is inclusive and accommodating to caregivers. As Ciara (Research Assistant, children) argued, *'that's totally in the hands of who your line manager is, and I'm sure that people have had different experiences if they're working on different types of projects where there are different deadlines and priorities and stuff'*.

Similarly, Marcus explained:

> I still think there is informal contracts that hold in each research groups, so it depends on whether your professor is a b*****d or not, basically whether it's acceptable to do flexible working. So, I still think those informal ways of arrangements within research groups will still be stronger than those actual policies.

Indeed, participants' narratives highlighted some significant variations at inter- and intra-institutional levels, which are maybe best encapsulated by the contrast between the following excerpts. Asked about the support received from her line manager, Pauline, for example, replied:

> He has no interest whatsoever I think, not me just any staff issues, not at all . . . people don't really want to hear it . . . . I mean some of my colleagues that I've worked with for a long time, they know and I know about their caring responsibilities, but no, it's not something we talk about at all.

In contrast, Sasha (Research Assistant, children and elderly parents) explained:

> [My manager] allowed me to be really flexible with my hours, so I can drop the girls off . . . they're happy for me to come and start at somewhere between half nine and ten. As I say they're flexible about my hours so I can do my one long day, so I only have to arrange childcare after school for one day a week. I can pick them on the other days, because I do 10 until two on those two short days. I can't fault them in terms of enabling the work–life balance that I think's really important to me . . . [They have been] really supportive, really, really good.

The recourse to individualised practices raises some important equity issues, both between academics with significant caring responsibilities and those without, and between carers. Our broader research in this area shows that policy-making processes and the effects of policies are much more complex than initially thought, with various levels of policy making interacting with each other, in ways which are rhizomatic rather than top down and causal (Deleuze and Guattari 1980). While the significance of the immediate working environment in the policy-making process may bring some benefits (for example, leading to a better understanding of the unit's culture and of the needs of individuals), access to resources becomes contingent on immediate colleagues and subject to their good will. In a HE context where line managers have significant discretion in the way they treat caregivers and other members of staff (Arksey 2002), this article suggests that this may result in a lack of consistency across and within line management lines. As a result, the translation of policies

into academic cultures which are 'carer-friendly' requires that awareness and understanding of care issues are facilitated at all levels of the institution and among all groups of staff, including those in line management positions. Findings from this article point to the intersectional ways in which care-related inequalities are formed and point to the need for further research, focusing for example on carers whose responsibilities and 'other identities' have attracted limited research and policy interests. Inequalities of access to resources and to a valued hyphenated identity as an academic carer are compounded by relationships of power based on gender, class, ethnicity, sexuality and dis/ability. While collegial practices are laudable, relying solely on individualised forms of support risks rendering care work invisible, ultimately perpetuating inequalities between those who can perform an academic identity and those who cannot.

**Author Contributions:** Writing–original draft, M.-P.M. and M.R.

**Funding:** This research was funded by the Leadership Foundation for Higher Education (2015/2016 Small Development Project Grant) and the Society for Research into Higher Education (2017 SRHE Research Awards).

**Acknowledgments:** We would like to thank the Leadership Foundation for Higher Education (now Advance HE) and the Society for Research into Higher Education (SRHE) for their support, as well as the members of staff who gave up some of their time to take part in the research.

**Conflicts of Interest:** The authors declare no conflict of interest.

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
