# Peer review of "‘You Scratch My Back and I’ll Scratch Yours’? Support to Academics Who Are Carers in Higher Education"

_socsci, doi:10.3390/socsci8060164_

Round 1

Reviewer 1 Report

Page 2, line 87, use the full name of SRHE in the first instance and for readers outside the UK who may not be familiar.

Page 3, line 94, would be good to include some contextual detail about Russell Group and post-1992 institutions as this does not easily translate for international readers.

Male and BME academics difficulties - more discussion and analysis of this difficulty would be good.

Page 3, line 134 - BME please use full name.

Page 5, line 220 - ‘the department is fine, it’s the wider university organisation that’s the problem - felt that this was very interesting and important point and that the analysis provided could be extended and further developed. I felt you did this to some extend in your conclusion but that the paper would benefit from additional discussion on this point.

Page 5, line 233 - disproportionate amount of care; I felt that this claim needed to be supported by the data.

Page 8, line 373 - intensive parenting and a discourse of neoliberal academia; I thought that both these concepts needed further discussion and references. Perhaps this contextual work could come earlier in the piece, you mention scarcity of work around care in HE other than parenting earlier so perhaps these two ideas are linked.

Page 8, line 376 - care-free, I think you need to reference Lynch here. - Carelessness: A hidden doxa of higher education - Arts and Humanities in Higher Education, 2010

Page 8 381, hyphenated identity, this is an interesting concept and one you draw on consistently but I think it could be shaped, referenced and developed further.

An excellent paper and important work to build the care in HE field beyond parenting.

Author Response

Dear Reviewer 1,

Many thanks for your comments. We have found these very helpful and have amended the paper accordingly. We attach the file with track changes to this email.

As well as the specific amendments you required, we have tried to strengthen the introduction and the conclusion. We have also done some minor editing to address typographical mistakes and to improve clarity and readability.

Again, many thanks.

With best wishes,

Professor Marie-Pierre Moreau

Reviewer 2 Report

Comments

1.    The introduction could communicate more efficiently what the study is about. What is the exact contribution of the paper to the (international) literature?

2.     The empirical context of the study should be motivated in the introduction.

3.    An alternative theoretical framework for the interpretation of the estimation results is based on the theory of compensating wage differentials (Bockerman et al. 2011). This issue should be mentioned in the revised version of the paper. 

4.    Are the interviews fully documented?

5.    The relevant limitations of the empirical approach should be stated in the concluding section of the paper.

6.    The concluding section of the paper should discuss more about the practical policy lessons that can be drawn from the interviews. 

Reference

Bockerman, P., Ilmakunnas, P. & Johansson, E. (2011). Job security and employee well-being: Evidence from matched survey and register data. Labour Economics, 18:4, 547-554.

Author Response

Dear Reviewer 2,

Many thanks for your comments - we have amended the paper accordingly. We attach the file with track changes to this email.

As well as the specific amendments you required, we have tried to strengthen the introduction and the conclusion. However, we were not able to include Bockerman et al. (2011) as the link with our article was unclear. Instead, we have included one additional reference (Lynch, 2010) as we acknowledge that this is a very influential piece of work.

We have also done some minor editing to address typographical mistakes and to improve clarity and readability.

Again, many thanks.

With best wishes,

Professor Marie-Pierre Moreau